# Integration of Antifouling and Underwater Sound Absorption Properties into PDMS/MWCNT/SiO_2_ Coatings

**DOI:** 10.3390/biomimetics7040248

**Published:** 2022-12-18

**Authors:** Pan Cao, Huming Wang, Mingyi Zhu, Yifeng Fu, Chengqing Yuan

**Affiliations:** 1College of Mechanical Engineering, Yangzhou University, Yangzhou 225127, China; 2School of Automobile and Traffic Engineering, Jiangsu University, Zhenjiang 212013, China; 3National Engineering Research Center for Water Transportation Safety, Reliability Engineering Institute, Wuhan University of Technology, Wuhan 430063, China; 4School of Transportation and Logistics Engineering, Wuhan University of Technology, Wuhan 430063, China

**Keywords:** integrated nanocomposite coating, surface characterization, environmentally-friendly coatings, antifouling performance

## Abstract

Any surface immersed in sea water will suffer from marine fouling, including underwater sound absorption coatings. Traditional underwater sound absorption coatings rely heavily on the use of toxic, biocide-containing paints to combat biofouling. In this paper, an environmentally-friendly nanocomposite with integrated antifouling and underwater sound absorption properties was fabricated by adopting MWCNTs-COOH and SiO_2_ into PDMS at different ratios. SEM, FTIR and XPS results demonstrated MWCNTs were mixed into PDMS, and the changes in elements were also analyzed. SiO_2_ nanoparticles in PDMS decreased the tensile properties of the coating, while erosion resistance was enhanced. Antibacterial properties of the coatings containing MWCNTs-COOH and SiO_2_ at a ratio of 1:1, 1:3, and 1:5 reached 62.02%, 72.36%, and 74.69%, respectively. In the frequency range of 1500–5000 Hz, the average sound absorption coefficient of PDMS increased from 0.5 to greater than 0.8 after adding MWCNTs-COOH and SiO_2_, which illustrated that the addition of nanoparticles enhanced the underwater sound absorption performance of the coating. Incorporating MWCNTs-COOH and SiO_2_ nanoparticles into the PDMS matrix to improve its sound absorption and surface antifouling properties provides a promising idea for marine applications.

## 1. Introduction

Underwater sound absorption coatings are composed of a polymer matrix and fillers, which are glued to the ship or submarine hull for absorbing underwater noise or active sonar. However, all surfaces that are subjected to sea water will be colonized by marine biofouling, which is the harmful accumulation of marine microorganisms in submerged parts of ships or marine facilities [1,2,3]. Inevitably, underwater sound absorption coating will also be impacted by biofouling. The growth of the organisms on underwater sound absorption coatings will not only roughen the surface, but also change the acoustic property of the coating—the underwater sound absorption performance will decrease. Traditional underwater sound absorption coatings need to be covered with toxic antifouling paints [4], which pose a serious threat to marine ecosystems and limit the underwater sound absorption performance. Therefore, it is necessary to develop one nanocomposite with integrated antifouling and underwater sound absorption properties. 

Polymers are commonly used for underwater sound-absorbing materials because of the similar acoustic impedance with water to reduce the reflection of incident sound waves [5]. Among these polymers, polydimethylsiloxane (PDMS), an artificial polymer, possesses stable physical and chemical properties and environmental friendliness [6], making PDMS aa widely used matrix for underwater sound absorption materials. Further, different structures or fillers are embedded into the PDMS matrix to dissipate the sound waves. Luo et al. [7] prepared interpenetrating polymer networks (IPN) by mixing PU/EP/UPR and polydimethylsiloxane (PDMS) materials in different proportions, and it was proven that IPN possessed excellent sound absorption performance. Gu et al. [8] reported the sound absorption performance of the composite coating of PU and CNTs, and the agglomeration of CNTs significantly enhanced the sound absorption performance of the coating. Nanoparticles are often used as fillers to enhance the mechanical properties and antifouling properties of polymers. Cavas et al. [9] modified PDMS with p-MWCNTs, carboxylate-functionalized MWCNTs and graphene oxide (GO) to improve the mechanical properties of PDMS. The performances of different dispersion methods and content of nanoparticles on the mechanical properties of PDMS were investigated, which confirmed that the addition of MWCNTs improved the tensile properties and the elongation of PDMS. Further, many studies reported that the addition of CNTs significantly improved the underwater sound absorption performance of PDMS materials [10,11]. Selim et al. [12] added TiO_2_ nanoparticles into the PDMS matrix as filler to fabricate a novel PDMS composites material with an excellent self-cleaning ability when the content of nanoparticles was 0.5%, which mechanical properties were determined by dispersion properties of nanoparticles. Jena et al. [13] produced a graphene oxide-nano-SiO_2_-polydimethylsiloxane composite coating on carbon steel through an anodic electrophoretic deposition combined with dip coating; the composite coating possessed excellent antibacterial activity and antibiological contamination performance due to the moderate toxic effect of GO and the presence of Si-O-Si. Meanwhile, the mixture of SiO_2_ and GO also enhances the corrosion resistance of the coating.

PDMS also has low surface energy. When fouling organisms attach to PDMS, the bonding strength between fouling organisms and PDMS is commonly very low; thus, fouling organisms can be easily detached by the shear force of water flows. Therefore, PDMS has been widely studied in recent years as a kind of silicone-based fouling-release coating (FRC) [14,15]. However, the poor performance in terms of mechanical properties, resistance to mechanical damage and antifouling properties on static substrate surfaces has greatly limited the application of PDMS in marine environments [16,17,18]. Many studies have tried to overcome the shortcomings of PDMS materials through chemical modification. In this study, SiO_2_ is used to modify PDMS-based nanocomposites. There have been a lot of studies on its application in the field of antifouling. Deng et al. [19] successfully prepared environmentally-friendly, efficient, low-cost Ag@TA-SiO_2_ nanoparticles with bactericidal properties through simple free radical copolymerization. Wang et al. [20] blended unmodified or PTM-grafted SiO_2_ nanoparticles and immersed them in ASW to prepare nanocomposite coatings, and developed an underwater superhydrophobic antifouling coating that could be repaired by seawater induction. Bhushan et al. [21] prepared superhydrophobic coatings on PET substrates by dipping hydrophobic SiO_2_ nanoparticles and methyl phenyl silicone resin.

As far as we know, only a few studies have been reported to improve the sound absorption materials’ antifouling properties [22]. In our previous study [10], it has been proven that PDMS/MWCNTs-COOH has good underwater sound absorption performance. In this study, SiO_2_ is introduced to for increasing antifouling property for underwater application without sacrificing the underwater sound absorption performance. Additionally, their mechanical, chemical, and physicochemical properties have been thoroughly investigated. The advantages of the novel coatings are as follows.(1)The coatings have both underwater sound absorption and antifouling properties, which will fill the gap in this filed.(2)Without adding any environmentally-unfriendly fungicide into the sample, we prepare non-toxic sound-absorbing and antifouling coatings.(3)In this study, we tested the tensile properties and water erosion resistance of the coatings to prove its durability.

## 2. Experimental Section

### 2.1. Materials

PDMS (Part A, Part B) was purchased from Dow Corning (Sylgard 184, Midland, MI, USA). Carboxylic Multi-Walled Carbon Nanotubes (MWCNTs-COOH, >95%; outside diameter: 10–20 nm; length: 10–30 μm) was purchased from Aladdin Biochemical Technology Co., Ltd. Dispersion-aiding additives based on solvent-free acrylate copolymers (Disperbyk-191 and 192) were supplied by BYK USA Inc. Silicon dioxide (SiO_2_, purity 99.5%, 30 ± 5 nm) was purchased from Shanghai Macklin Biochemical Co., Ltd. 2216E culture medium was purchased from Qingdao Hope Bio-Technology Co., Ltd. (Qingdao, China). Ethanol absolute (≥99.7%) was acquired from Sinopharm Group Chemical Reagent Co., Ltd. (Shanghai, China). Glutaraldehyde (purity 50%) was purchased from Aladdin Biochemical Technology Co., Ltd. (Shanghai, China). All chemicals were used without further purification.

### 2.2. Fabrication of Nanocomposite Material

MWCNTs-COOH and SiO_2_ were added to modify PDMS. Four groups with different ratios are shown in Table 1 (PDMS without mixing MWCNTs and SiO_2_, and PDMS with different ratios of MWCNTs and SiO_2_ of 1:1, 1:3, and 1:5, respectively) and denoted as PDMS, PCSi1, PCSi3 and PCSi5.

The schematic illustrations for the detailed sample fabrication procedures are shown in Figure 1. MWCNTs-COOH, SiO_2_, and Disperbyk-191 and 192 were mixed in proportion and stirred with PDMS (Part A). To improve the dispersion of MWCNTs-COOH and SiO_2_ in the PDMS, the mixed liquid was ground five times on a three-roll mill. The distance between the finish roller was set at 20 μm, and the distance between the coarse roller changed from 100 to 20 μm. To prevent sample spilling, the instrument was started at 65 rpm and then slowly accelerated to 100 rpm until the sample was evenly distributed over the rollers. The whole process took approximately 30 min. The mixture was combined with PDMS (Part B) after being ground on the three-roll mill and then degassed in a vacuum drying chamber. This process lasted for 30 min, then the sample was poured into the mold and degassed again for 5–10 min to further remove incorporated air bubbles. After degassing, the mold and liquid samples were placed in the oven and solidified at 65 °C for 90 min. A solid sample was obtained. 

### 2.3. Morphology and Chemistry Characteristics

The surface micromorphology was observed by a field emission scanning electron microscope (FE-SEM, Gemini SEM300, Oberkochen, Germany). The surface compositions were collected by a X-ray photoelectron spectrometer (XPS, ESCALAB 250Xi, Waltham, MA, USA). Fourier transform infrared spectrometer (FTIR, Cary 610/670, Palo Alto, CA, USA) was applied to collect the information of PDMS and PDMS-based nanocomposite materials in the scanning range from 4000 to 400 cm^−1^ for 128 scans at a spectral resolution of 4 cm^−1^. The roughness was characterized by a 3D optical profilometer (Contour GT-K, Bruker, Billerica, MA, USA). The water contact angle of the samples’ surfaces was evaluated by a contact angle measuring instrument (JC2000D, Shanghai, China), three points are taken for each sample surface measurement.

### 2.4. Mechanical Property Test

#### 2.4.1. Tensile Test

The mechanical properties were tested through a tensile testing machine (WDW-100) and the experimental procedure is shown in Figure 2. In the test, a static force measuring element of 1000 N was used to apply tensile force to the sample at a crosshead speed of 90 mm/min. The tensile test samples were rectangular prisms of 60 mm length, 20 mm width, and 5 mm height. Three samples were tested for each condition. The load and displacement of each test were recorded.

#### 2.4.2. Water Erosion Test

A water erosion test machine was used for the water erosion test to study the influence of the nanoparticles on the erosion resistance of the coating. Sand water with a sediment content of 12% was used for erosion with an erosion angle of 60° and erosion speed of 12 m/s. The samples were eroded for 12 h, during which time the samples were taken out every 3 h for weighing and recording the surface topography after 3–5 min cleaning in an ultrasonic cleaning instrument. The mass loss (*M*) and mass loss rate (*M_r_*) of materials were calculated by the following equation.
*M* = (*m_i_* − *m*_0_)/*A*(1)
*M_r_* = *M*/*t*(2)
where *m_i_* is the mass of the sample after erosion, and *m*_0_ is the initial mass of the sample. *A* is the scouring surface area of the sample, and *t* is the test time.

### 2.5. Antifouling Test

#### 2.5.1. Bacterial Culture

*Vibrio natriegens (V. natriegens)*, a common bacterium in East China Sea waters, was selected as the fouling organism to evaluate the antifouling performance of the samples. The 2216E culture medium was placed into a vertical autoclave for 30 min (more than 15 min at 121 °C). After sterilization, the medium was poured into the Petri dish (diameter: 9 cm) for generating 2216E solid medium. *V. natriegens* was inoculated on 2216E Agar culture plate, and placed in an incubator (37 °C) for 24 h invertedly. The single colony of *V. natriegens* was selected into a tube with 6 mL 2216E culture medium and cultured under a constant-temperature shaker (37 °C, 130 rpm) until the medium became turbid.

#### 2.5.2. Plate Counting

The bacterial solution was diluted 100 fold and the samples were placed into 24-well cell culture plates with 1 mL diluted bacterial solution per well. Culture plates with samples were placed into a constant-temperature shaker (37 °C, 60 rpm) for 24 h. After removing residual liquid, the samples were cleaned with 8.5 g/L sodium chloride (NaCl) solution. Then, the samples were put into a centrifuge tube with 3 mL NaCl solution, and centrifuged for 20 min in a centrifuge (4000 rpm). The centrifuged liquid was diluted 1000 fold, then 100 μL drops were injected on Petri dish with 2216E solid medium and were spread evenly. The bacteria were cultured invertedly in a constant-temperature incubator (37 °C) for 12 h, and the bacteria on the Petri dish were counted.

#### 2.5.3. FE-SEM Observation

The monoclonal *V. natriegens* were picked to obtain a concentration of 1 × 10^6^ CFU/mL bacterial solution (37 °C, 180 rpm, 48 h) in 2216E medium. 304 SS, PDMS and nanocomposite coating were cultured in *V. natriegens* solution (37 °C, 48 h), and the treated samples were then washed with sterile PBS solution to remove unadhered or loosely attached bacteria. The samples were cleaned with PBS solution and treated with in 2.5% glutaraldehyde at 4 °C for 4 h for fixing bacteria onto a solid surface. Bacteria dehydration was carried out using 25, 50, 75, 90, and 100% ethanol solutions (10 min/concentration) in turn and then drying under vacuum for FE-SEM observation.

### 2.6. Underwater Sound Absorption Test

Underwater sound absorption properties were tested in a water-filled impedance tube. At least 24 h before the tests, the tube was filled with distilled water to remove all air bubbles (room temperature). Samples were installed on the top of the tube with air as the soft backing. One-dimension plane waves were generated by the underwater transducer, and then the sound waves were reflected by the sample. Two hydrophones were enlisted to record the sound signal. The sound absorption coefficient can be calculated from the reflection coefficient ®. The working frequency of the water-filled impedance tube is from 500 to 5000 Hz.
*α* = 1 − |*r*|^2^(3)

## 3. Results

### 3.1. Morphology and Chemistry Characteristics

FE-SEM, FTIR, and XPS were performed to check the dispersion of MWCNTs-COOH and SiO_2_ in the PDMS matrix. Figure 3 demonstrated the SEM images of the surfaces and cross-sections of nanocomposite samples. The images of PDMS showed that the surface and interior of the sample are clean and there is no other material except the PDMS matrix (Figure 3A-1,B-1). While it can be clearly observed that CNTs and silica nanoparticles were incorporated into PDMS after treatment. Some visible tubular MWCNTs appeared on the PDMS matrix, and most of them were evenly dispersed, while some MWCNTs were clustered together (Figure 3A-2,A-3,A-4,B-2,B-3,B-4). Both single and agglomerated MWCNTs can be observed on the cross-sections of the samples because most of the MWCNTs-COOH were gathered inside of the sample rather than on the surface of the sample. Many silica nanoparticles can also be found clearly from cross-sectional images. The SEM images indicated that MWCNTs-COOH and SiO_2_ nanoparticles were adopted and well mixed into the PDMS matrix. Apart from this, the surface profile of samples (Appendix A) showed that nanoparticles changed the morphology of the coating surface. The roughness Ra of different surfaces changed slightly, but the roughness of different positions of the same sample varied significantly (Appendix A). There was a large height drop on the coating surface due to the slight fluidity of liquid sample during curing. The uneven distribution of nanoparticles during mixing led to the surface profile difference throughout the coating. The slight variation of roughness changes the contact angle of samples’ surface as shown in Appendix A, and contact angle of PDMS, PCSi1, PCSi3 and PCSi5 were 111.6° 108.7°, 110.2°, 113.3°, respectively.

FTIR was used to collect the element information and functional groups of MWCNTs-COOH and SiO_2_-treated PDMS, and Figure 4A–D showed FTIR spectra of PDMS, PCSi1, PCSi3 and PCSi5, respectively. It can be observed that the addition of MWCNTs-COOH and SiO_2_ has a slight influence on the main functional groups in the sample. As shown in Figure 4A–D, the peaks that appeared at 785–815 cm^−1^ were derived from -CH_3_ rocking and Si-C stretching in Si-CH_3_. The two peaks at approximately 1055–1090 cm^−1^ were Si-O stretching of Si-O-Si, and the symmetric -CH_3_ deformation peak of Si-CH_3_ can be observed between 1245–1270 cm^−1^, while asymmetric -CH_3_ stretching in Si-CH_3_ corresponded to the peaks at approximately 2950–2970 cm^−1^ [23]. Figure 4E is the comparison of FTIR spectra of the nanoparticle-treated samples. It is obvious that the intensity of the Si-O-Si peak increased with the increasing content of SiO_2_. The quantitative analysis of FTIR can be carried out by the Beer–Lambert law [24,25].

The mathematical expression of Beer–Lambert law is:*A* = *εlc*
(4)
where *A* is absorbance; *ε* is the molar absorption coefficient, which is related to the wavelength of incident light and the properties of the measuring object; *l* is path length (e.g., the thickness of the sample in the assay); *c* is concentration.

In the quantitative analysis of Si-O-Si peaks in the four groups of samples, the wavelength of incident light (*ε*) and the thickness (*l*) of the infrared spectrum detection of the three samples are constant, which means the concentration of the Si-O-Si peak *c* is proportional to the absorbance *A*. The intensity of Si-O-Si peak of nanoparticle-treated samples increased with the increase in SiO_2_ content, which was because that each Si atom in PDMS is connected by two O atoms and one C atom, whereas in SiO_2_ each Si atom is connected to all four O atoms. The functional groups different from PDMS proved the existence of MWCNTs-COOH and SiO_2_ in PDMS.

XPS was used to further determine the element’s chemical information of C, O, and Si in the four group samples (Figure 5). The specific contents of different elements in the four group samples are shown in Table 2. The mass proportions of C, O, and Si in the PDMS were 32.14%, 26.69%, and 41.17%, and the mass ratio C/O and C/Si were 1.20 and 0.78, respectively. After the addition of MWCNTs-COOH and SiO_2_ at a ratio of 1:1, the mass proportion of C and O increased to 33.34% and 27.06%, while Si decreased to 39.06%, and so the mass ratio C/O increased to 1.23, and the mass ratio C/Si increased to 0.84. Subsequently, with the increasing content of SiO_2_, the contents of Si and O increased to 40.11% and 40.53%, 27.08%, and 28.51%, respectively, while the content of C decreased to 32.81% and 30.96%. Correspondingly, the mass ratios of C/O in the samples decreased to 1.21 and 1.08, while the mass ratios of C/Si dropped to 0.82 and 0.76. However, the contents of Si in PCSi1, PCSi3 and PCSi5 were lower than that in the pure PDMS due to more silicone-free addition. It can be seen from Table 1 that the content of PDMS is reduced by 3% after the addition of 1:1 MWCNTs-COOH and SiO_2_. Although the content of SiO_2_ increases by 0.5%, other silicone-free components increased including disperbyk-191 and 192, and MWCNTs-COOH, which led to decrease in the content of element Si. The mass proportion of Si in PDMS was 41.18%, which was less than that in SiO_2_ (46.67%), the content of Si in PCSi3 and PCSi5 will be improved. The theoretical Si contents in PDMS, PCSi1, PCSi3 and PCSi5 were 41.18%, 40.17%, 40.23%, and 40.28%, respectively. The XPS results were very close to the theoretical values, which indicated that the MWCNTs-COOH and SiO_2_ were adopted into PDMS successfully. The XPS spectra were consistent with EDS results (Appendix A). It was found that the contents of Si in PCSi1, PCSi3 and PCSi5 were lower than PDMS, while C, O and Si were uniformly distributed on the surface of the coatings without significant changes in content (Appendix A). The variation of SiO_2_ contents in PCSi1, PCSi3 and PCSi5 had slight influence on the content of elements since the composition of PDMS dominated the composite coatings.

### 3.2. Mechanical Property Test

#### 3.2.1. Tensile Test

The tensile test result is presented in Figure 6. It shows that ultimate tensile strengths (UTS) of pure PDMS and PCSi1 were both approximately 1.1 MPa. With the mass ratio of MWCNTs-COOH and SiO_2_ increasing to 1:3 (i.e., PCSi3), the UTS dropped to 0.6 MPa. Additionally, the UTS of PCSi5 was 0.7 MPa. In terms of Young’s modulus, pure PDMS samples had the highest Young’s modulus, and the change in SiO_2_ content did not affect the Young’s modulus of the other three groups of samples (i.e., PCSi1, PCSi3, and PCSi5). That is, the three groups of samples had the same Young’s modulus, which was smaller than that of pure PDMS. This indicates that the addition of MWCNTs-COOH and SiO_2_ nanoparticles adversely affects the Young’s modulus of PDMS materials, but the increase in SiO_2_ content did not affect the Young’s modulus of the nanocomposites.

#### 3.2.2. Water Erosion Test

Table 3 showed the water erosion results of the samples, and the surfaces of the samples were smooth before being flushed. Some scour marks appeared on the surfaces after suffering 3 h rinsing, and the scour mark became deeper and spread to the surrounding gradually with the scouring time. After 12 h water erosion, the trace spread almost to the entire surface of the samples. Sample’s weight also decreased with the increase in water erosion time, and the changes in sample weight are shown in Table 4. The mass loss and mass loss rate of the samples were calculated according to Formulas (1) and (2), respectively. The mass of PDMS was 1.957 g, and the mass loss was 82.84 g·m^−2^ after 12 h water erosion with a mass loss rate of 6.903 g·m^−2^·h^−1^ (Figure 7A). The mass of the PDMS matrix with MWCNTs-COOH and SiO_2_ at a ratio of 1:1 was 2.078 g, and the mass loss and mass loss rate were 112.42 g·m^−2^ and 9.368 g·m^−2^·h^−1^ after 12 h water erosion (Figure 7B). The mass of PCSi3 and PCSi5 were 2.002 g and 2.055 g, while the mass loss rate reached 6.410 and 1.973 g·m^−2^·h^−1^ after 12 h water erosion, which resulted from 76.92 and 23.67 g·m^−2^ mass loss (Figure 7A,B).

### 3.3. Antifouling Test

After 24 h co-culture with bacterial medium, FE-SEM and plate counting were conducted for evaluating the antibacterial performance of nanocomposite coatings and the results are shown in Figure 8. Figure 8A showed FE-SEM images of 304 stainless steel and PDMS containing MWCNTs-COOH and SiO_2_ at a ratio of 1:1, 1:3, and 1:5 after incubation. There were dense visible bacteria on 304 stainless steels after 24 h of incubation, fewer bacteria could be observed with the addition of more SiO_2_ and it was difficult to find bacteria on the sample surface if 2.5 wt% SiO_2_ was added. Figure 8B demonstrated plate counting results of 304 stainless steel and PDMS containing MWCNTs-COOH and SiO_2_ at a ratio of 1:1, 1:3 and 1:5 after incubation. The number of bacteria adhered on 304 stainless steel was significantly higher than that of PDMS-based material, and the bacterial concentrations of the sample surface eluents are shown in Figure 9. The bacterial concentration of 304 stainless steel surface eluent was 2.58 × 10^6^ CUF/mL, while the values decreased to 9.80 × 10^5^ CUF/mL, 7.13 × 10^5^ CUF/mL and 6.53 × 10^5^ CUF/mL for PCSi1, PCSi3 and PCSi5, with antibacterial efficiencies of 62.02%, 72.36% and 74.69%, respectively. 

### 3.4. Underwater Sound Absorption Test

The underwater acoustic property was tested by a water-filled impedance tube, and the results are shown in Figure 10. At the frequency range from 500 to 1000 Hz, the underwater sound absorption coefficients of the samples with different proportions of MWCNTs-COOH and SiO_2_ nanofillers (i.e., PCSi1, PCSi3, and PCSi5) were not significantly improved, especially below 750 Hz. When the frequency was higher than 1000 Hz, the underwater sound absorption coefficients of PDMS samples with nanofillers were significantly higher than that of pure PDMS. In this frequency range, the underwater sound absorption coefficient of pure PDMS was generally lower than 0.6, while the underwater sound absorption coefficients of the three groups of samples with nanofillers increased to 0.8 or above. Especially, the underwater sound absorption coefficient of PCSi5 was generally higher than that of PCSi1 and PCSi3. From the perspective of the full frequency range, pure PDMS had a low peak of underwater sound absorption coefficient between 500 and 1000 Hz, but the underwater sound absorption performance was weakened in other frequency range. However, the underwater sound absorption coefficients of PCSi1, PCSi3, and PCSi5 showed an upward trend between 500 and 1500 Hz. They tended to be stable when the frequencies were higher than 1500 Hz, and the underwater sound absorption coefficient remained at 0.8. Meanwhile, the underwater sound absorption coefficients of PCSi1, PCSi3, and PCSi5 all had a peak value within the frequency range of 1000–1500 Hz. Marine animals are sensitive to the sound frequency range, which can protect marine animals from noise in this frequency range well.

## 4. Discussion

In this study, MWCNTs-COOH and SiO_2_ nanoparticles were successfully added to the PDMS. With the increase in SiO_2_ content, Si-O-Si bonds in the samples also increased gradually. This explains the improvement of antifouling performance of the samples.

Baier curve was usually used to explain the relationship between surface properties and the amount of bio-organisms adhered on surface [9]. The amount of adhered bio-organisms on surface would be lowest when the critical surface free energy (SFE) reached 22~25 mN/m according to Baier curve, the SFE range was the fouling release area. The critical surface of PDMS can meet the conditions of the fouling release area [26], which gave PDMS excellent antifouling performance under dynamic conditions. The low surface energy of PDMS was determined by a large number of Si-O-Si bonds in the PDMS structure [13]. Brady and Singer [27] revealed that bio-adhesion was associated with (Eγc)1/2 positive correlation, where E is the elastic modulus and γc is the critical surface energy of the polymer. Adding nanoparticles reduced the elastic modulus of the material and the biological adhesion of the coating. Although the results showed that the improvement of antifouling performance is not obvious with the increasing in SiO_2_ amount, the non-toxic coating improved the antifouling performance of substances by reducing surface bio-adhesion. Several different antifouling coatings were selected for comparison, as shown in Table 5. In contrast, PCSi5 has lower antibacterial efficiency compared with some other coatings. However, PCSi5 is easy to prepare as it possesses a single-layer structure but not containing additional bactericide triclosan. V. natriegens, a typical marine fouling organisms, was selected to evaluate the antifouling performance of PCSi5. On the other hand, PCSi5 not only had satisfactory antifouling performance, but also had excellent underwater sound absorption performance, which was not covered by other studies.

The sound absorption performance of the new coating is significantly improved. According to previous studies, after adding MWCNTs-COOH and surfactants into PDMS material, the underwater acoustic absorption coefficient of the material can be increased to more than 0.75 above 1500 Hz [16]. In this experiment, after the addition of SiO_2_, the underwater sound absorption coefficient of the material was not significantly improved above 1500 Hz. Only when 2.5% SiO_2_ was added did the sound absorption coefficient of the material slightly increase to more than 0.8. It can be concluded that MWCNTs-COOH contributed to most of the improvement of the underwater sound absorption performance of the materials. The improvement of the underwater sound absorption performance of MWCNTs-COOH materials can be attributed to the relative displacement of MWCNTs-COOH nanoparticles in the PDMS matrix under the action of acoustic waves and the dissipation of acoustic energy under the action of friction [17]. This is probably the reason why relatively more SiO_2_ nanoparticles improved the sound absorption performance of the material. In addition, during the addition of MWCNTs-COOH, some air microbubbles were introduced into the matrix, and the sound waves were dissipated when passing through these microbubbles [28]. In addition, the peak of the sound absorption coefficient of materials near 1000 Hz can also be explained by the resonance theory [29,30]. The experimental sample was a flexible nanocomposite material, which can be treated as a spring-mass system. When the frequency of the incident sound approached the natural frequency of the material, a peak of sound absorption was formed.

In addition, the poor mechanical properties of PDMS also limit its application. The influence of nanofiller on the tensile properties of the samples is very complicated. The main factors affecting the tensile properties of nanocomposites are the distribution of nanoparticles across the matrix and the transfer form of load on the matrix caused by the overall bonding strength between polymer and nanoparticles [31,32]. On the one hand, nanoparticles will produce some voids in the polymer matrix, resulting in the reduction in Young’s modulus. On the other hand, the strong bonding between nanoparticles and polymers will increase the young’s modulus. However, nanoparticles at high content will cause agglomeration and stress concentration, which have an adverse impact on the tensile properties of the coating. In this study, when less SiO_2_ was added, the SiO_2_ nanoparticle was unevenly distributed and voids were generated in the PDMS matrix, resulting in the fracture of the composite under even small stress. In addition, the strong covalent bond [33] formed between MWCNTs-COOH and PDMS made positive effect on the UTS of the material. Therefore, when MWCNTs-COOH and SiO_2_ at a ratio of 1:1 were added, the UTS of the material was similar to that of pure PDMS. When 1.5% SiO_2_ was added, the UTS of the composite was the smallest and the tensile properties are the worst. In the study of Bahramnia et al. [34], a hybrid polymer nanocomposite coating containing epoxy resin/polyurethane mixture, MWCNTs, and SiO_2_ nanoparticles were prepared, and the tensile test was investigated. The optimum SiO_2_ content was 0.75%. In this study, it can be predicted that when the content of SiO_2_ increases (more than 2.5%), the UTS of the nanocomposite will also increase, and then decrease after reaching an optimal content. The optimum content of SiO_2_ can be supplemented by subsequent research. Meanwhile, the change in mass loss indicated that the water erosion resistance of the surfaces increased with the addition of MWCNTs-COOH and SiO_2_. It was worth noting that the water erosion resistance of the samples became stronger with the increase in SiO_2_ content, and erosion resistance after 12 h water erosion was significantly lower than that of PDMS. The erosion resistance of materials is influenced by many factors, including the reinforcement materials type, combination ratio of matrix and reinforcement materials, expansion rate, and the hardness of materials [33]. As a carbon-based material, the uniform distribution of CNTs in nanocomposites has a great influence on the erosion resistance of the materials. Herein, mass loss of PCSi3 increased after adding MWCNTs-COOH and SiO_2_, which due to uneven dispersion of MWCNTs-COOH as filler, resulting in negative erosion resistance effects. SiO_2_ has a positive effect on the erosion resistance of the material, which may be explained by the surface property of the material. The visible texture of the material surface became blurred with the increase in SiO_2_, and the texture of surface became completely invisible for 2.5% SiO_2_ added PDMS, which indicated that SiO_2_ was wrapped in PDMS. SiO_2_ made the material more compact and has better erosion resistance capacity under the encapsulation of PDMS, then the water containing sand was not able to enter into PDMS and caused surface crack. In addition, SiO_2_ may also compensate for some of the material’s flaws by creating a thin protective layer. The mass loss rate of pure PDMS is 6.92 g·m^−2^·h^−1^, while that of PCSi5 is 2.02 g·m^−2^·h^−1^. This represents a 70.8% increase in durability of PCSi5 compared to pure PDMS.

**Table 5 biomimetics-07-00248-t005:** Antifouling performance of different antifouling coatings.

Number	Components	Bacteria	Antibacterial Efficiency	Characteristics	References
1	PDMS/MWCNTs-COOH/SiO_2_	*V. natriegens*	74.69%	Environmental friendly Simple structure	/
2	NH_2_-UiO-66/NH_2_-PDMS/epoxy resin	*E. coli*	79.42%	Best antifouling property Environmental unfriendly	[35]
3	NH_2_-UiO-66/NH_2_-PDMS/epoxy resin/triclosan		>99.98%		
4	PDMS/HD-SiO_2_		37.8%	Complex structure	[36]
5	Composite coating with outer PDMS/PAA-ZnO and inner PDMS/HD-SiO_2_ layer		81.1%		

## 5. Conclusions

MWCNTs-COOH and SiO_2_-modified PDMS nanocomposites have been fabricated. A series of tests have been conducted to characterize the effects of filler type as well as filler concentration on the morphology, mechanical property, water erosion resistance, antifouling performance, and underwater sound absorption property of these new nanocomposites. The results indicated that the nanocomposite coating added with MWCNTs-COOH and SiO_2_ had relatively excellent antifouling performance. The antibacterial properties of the coatings containing MWCNTs-COOH and SiO_2_ at a ratio of 1:1, 1:3, and 1:5 reached 62.02%, 72.36%, and 74.69%, respectively. The modified PDMS nanocomposite coatings also maintained excellent performance in sound absorption. In particular, the average sound absorption coefficient of PDMS samples with the addition of MWCNTs-COOH and SiO_2_ at a ratio of 1:5 reached 0.9. The addition of SiO_2_ can improve the water erosion resistance of the material. Therefore, the composite coating can be more stable in marine applications because the addition of MWCNTs-COOH and SiO_2_ generally enhanced its erosion resistance. The modified nanocomposite coating has a broad application prospect in the marine antifouling and underwater sound absorption.

In this study, the tensile properties of the coatings can be further improved. Therefore, we plan to find new methods to improve the tensile properties of the coating in future work. Optimization of the poor mechanical properties of nanocomposite coatings is a promising work. In addition, we noticed that the antibacterial rate of the coatings can also be improved, and we plan to use more nanofillers in future work to further improve the antibacterial rate of the coatings.

## Figures and Tables

**Figure 1 biomimetics-07-00248-f001:**
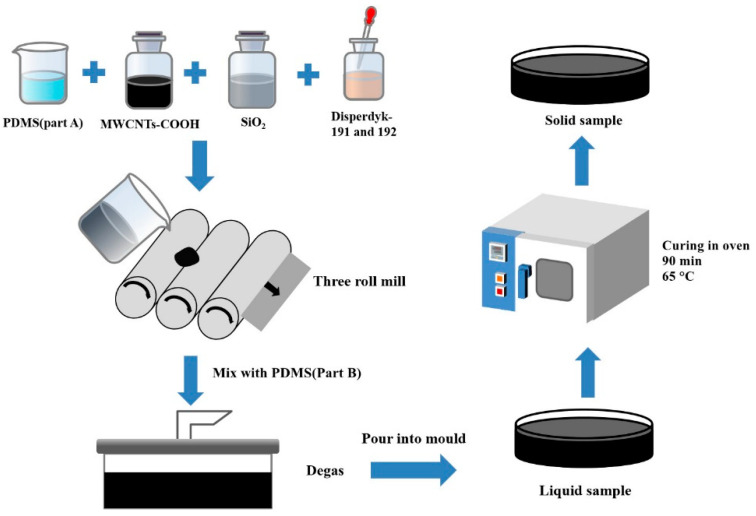
Schematic illustration of the fabrication procedures.

**Figure 2 biomimetics-07-00248-f002:**
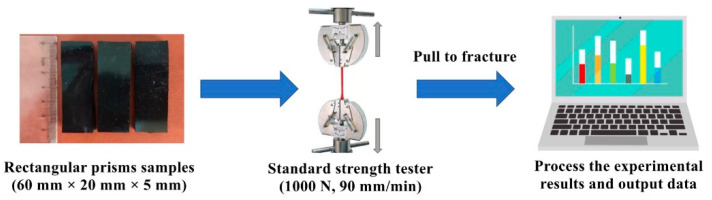
Schematic illustration of tensile test.

**Figure 3 biomimetics-07-00248-f003:**
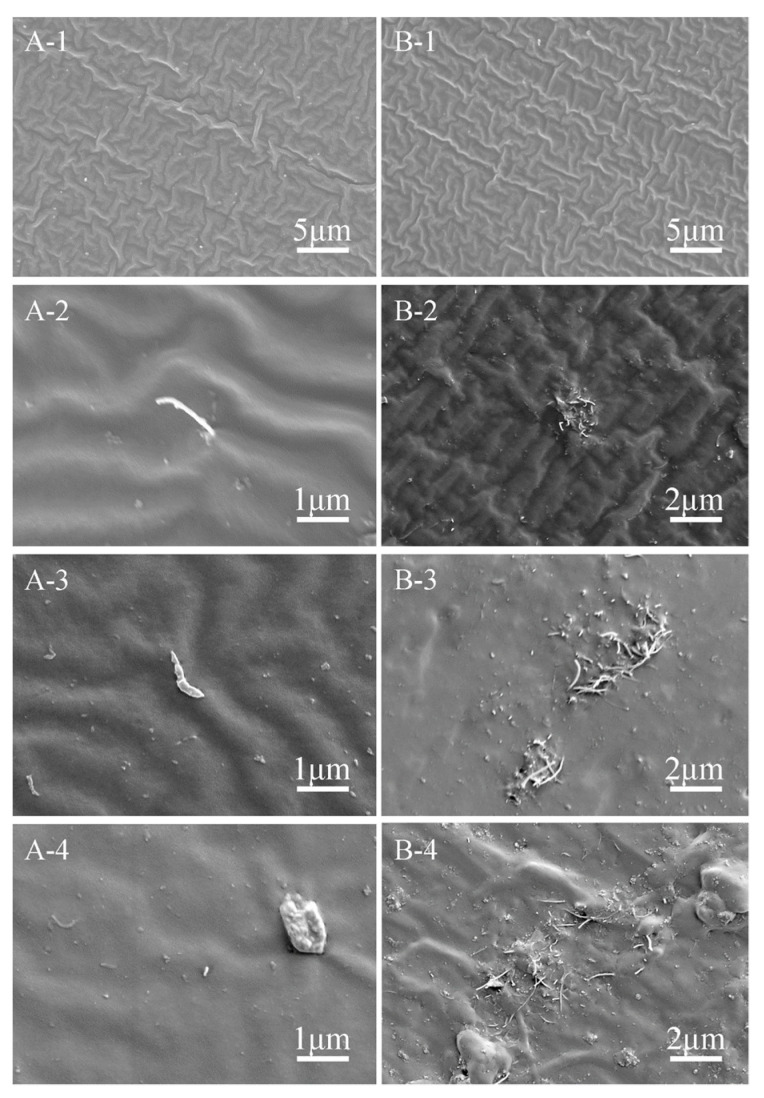
SEM images of the surfaces (**A**) and cross-sections (**B**); -1: pure PDMS; -2: PCSi1; -3: PCSi3; -4: PCSi5.

**Figure 4 biomimetics-07-00248-f004:**
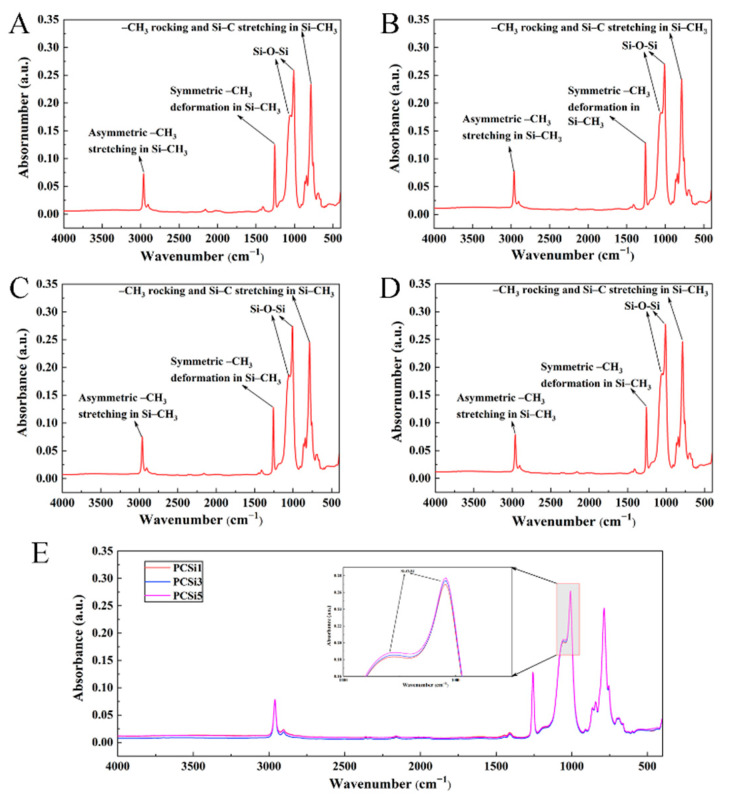
FTIR spectrum of pure PDMS (**A**), PCSi1 (**B**), PCSi3 (**C**), PCSi5 (**D**) and comparison of MWCNTs-COOH and SiO_2_-treated samples (**E**).

**Figure 5 biomimetics-07-00248-f005:**
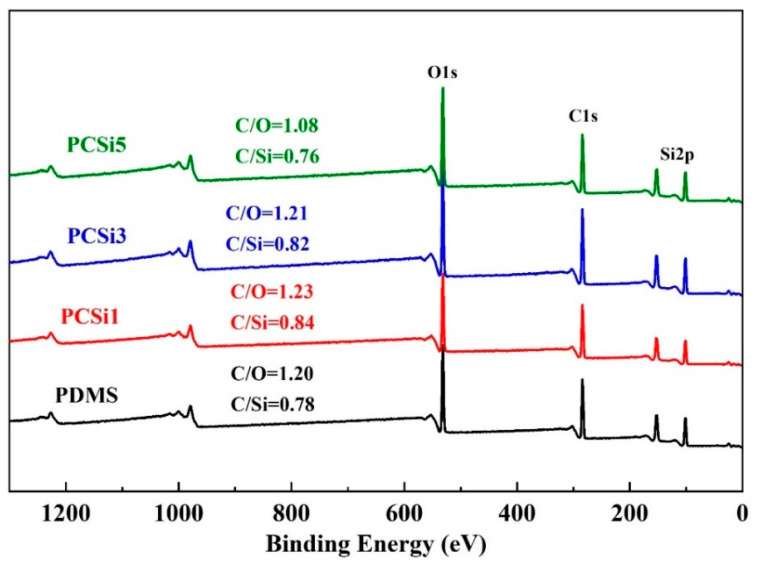
XPS spectra of different samples.

**Figure 6 biomimetics-07-00248-f006:**
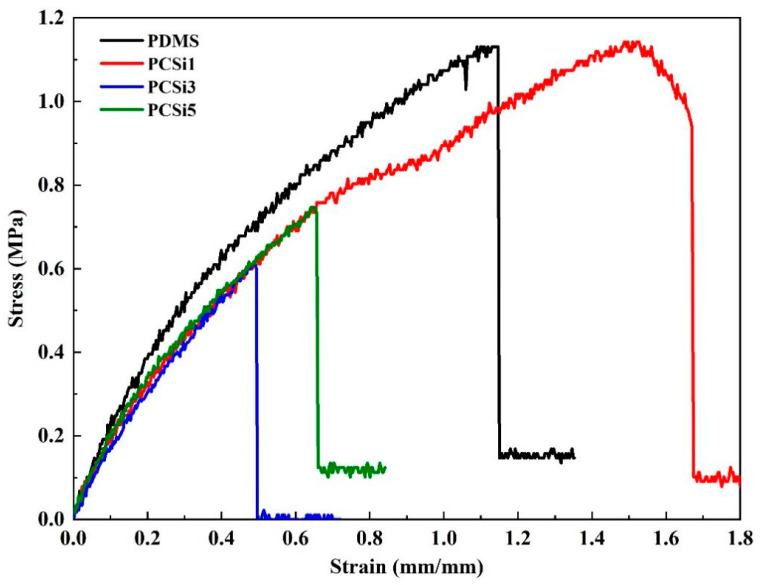
Stress–strain diagram for tensile test.

**Figure 7 biomimetics-07-00248-f007:**
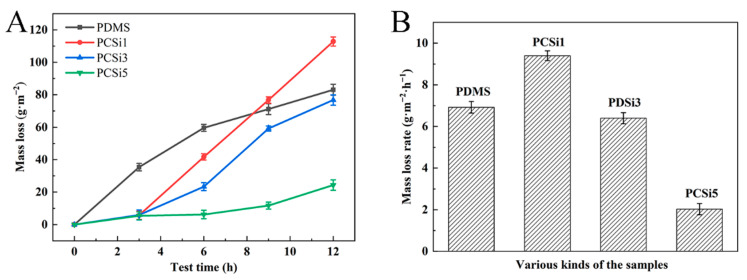
Mass loss versus time curve (**A**) and mass loss rate of the samples (**B**).

**Figure 8 biomimetics-07-00248-f008:**
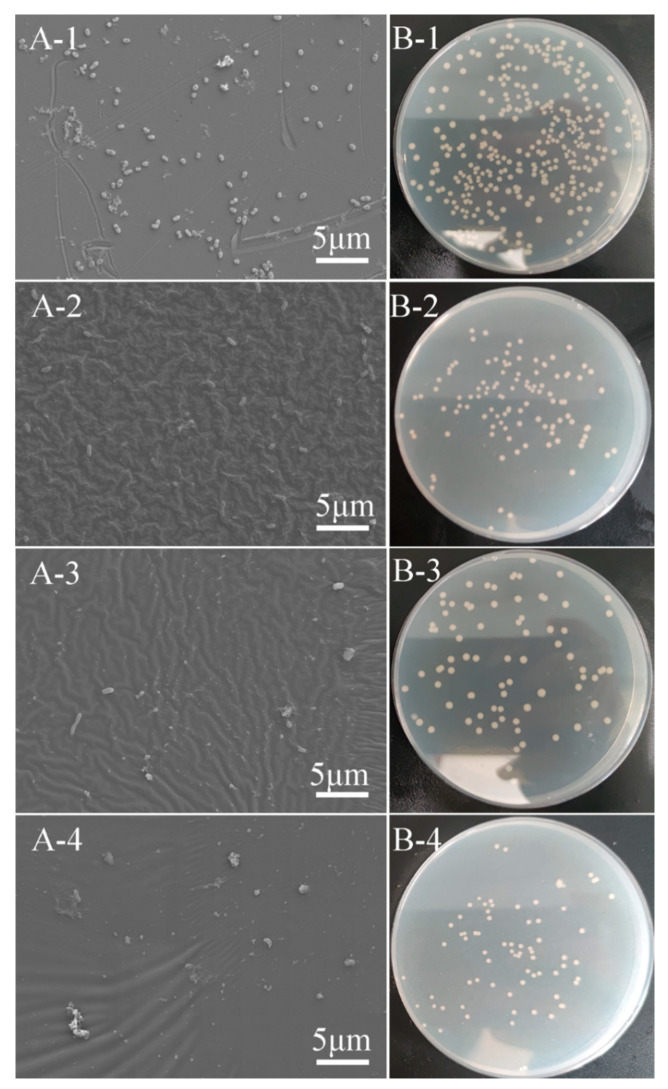
FE-SEM images (**A**) and plate counting (**B**) after the antifouling test; -1: 304 stainless steel, -2: PCSi1, -3: PCSi3, and -4: PCSi5.

**Figure 9 biomimetics-07-00248-f009:**
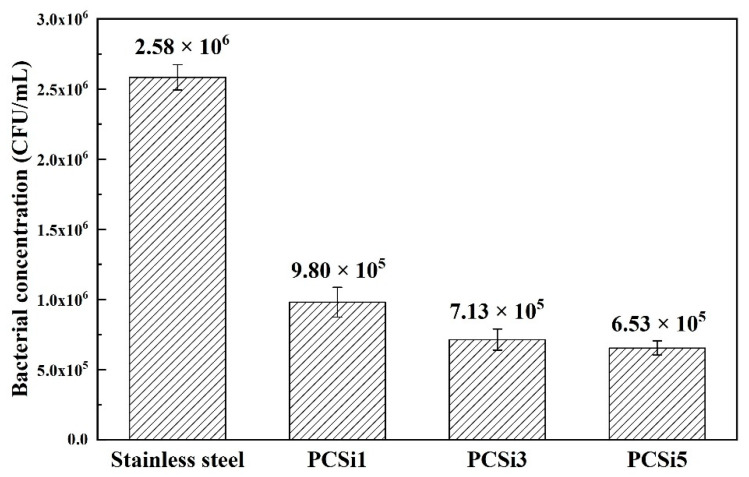
Plate counting statistics of samples after the antifouling test (error bar).

**Figure 10 biomimetics-07-00248-f010:**
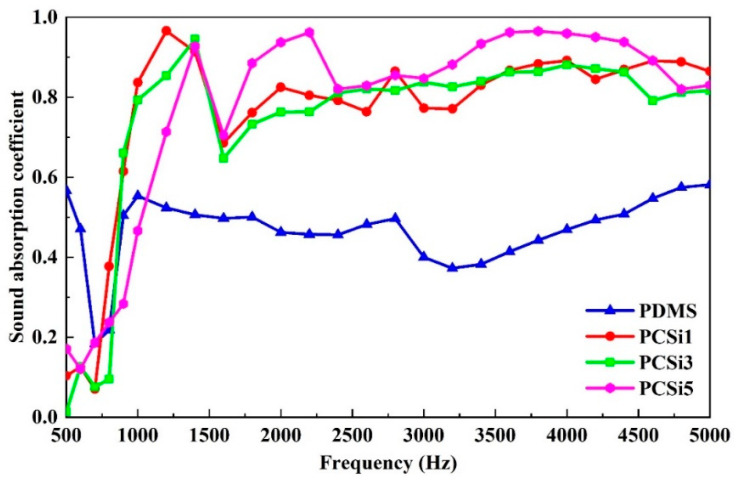
Underwater acoustic absorption performance of different samples.

**Table 1 biomimetics-07-00248-t001:** The compositions of different samples.

Name	PDMS	Disperbyk-191 and 192	MWCNTs-COOH	SiO_2_	Ratio
PCSi1	97%	2%	0.5%	0.5%	1:1
PCSi3	96%	2%	0.5%	1.5%	1:3
PCSi5	95%	2%	0.5%	2.5%	1:5
PDMS	100%	0%	0%	0%	N/A

**Table 2 biomimetics-07-00248-t002:** The contents of different elements in four group samples.

Sample	C		O		Si	
	wt%	Atomic%	wt%	Atomic%	wt%	Atomic%
PDMS	32.14	39.77	26.69	24.78	41.17	21.84
PCSi1	33.34	41.02	27.06	24.97	39.60	20.88
PCSi3	32.81	40.58	27.08	25.13	40.11	21.27
PCSi5	30.96	38.58	28.51	26.64	40.53	21.64

**Table 3 biomimetics-07-00248-t003:** Surface images after water erosion.

Samples	0 h	3 h	6 h	9 h	12 h
PDMS	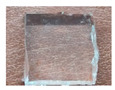	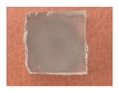	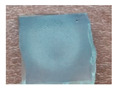	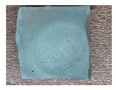	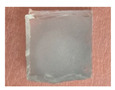
PCSi1	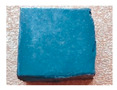	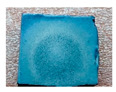	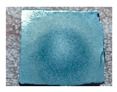	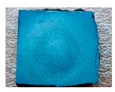	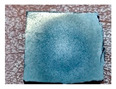
PCSi3	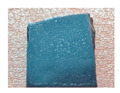	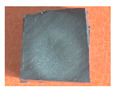	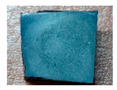	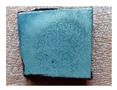	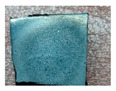
PCSi5	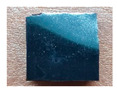	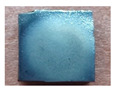	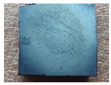	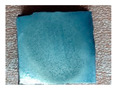	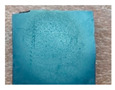

**Table 4 biomimetics-07-00248-t004:** Mass of samples after erosion.

Test Time	0 h	3 h	6 h	9 h	12 h
PDMS	1.957 g	1.951 g	1.947 g	1.945 g	1.943 g
PCSi1	2.078 g	2.077 g	2.071 g	2.065 g	2.059 g
PCSi3	2.002 g	2.001 g	1.998 g	1.992 g	1.989 g
PCSi5	2.055 g	2.054 g	2.054 g	2.053 g	2.051 g

## Data Availability

The data presented in this study are available on request from the corresponding author.

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
