# Peer review of "Integration of Antifouling and Underwater Sound Absorption Properties into PDMS/MWCNT/SiO2 Coatings"

_biomimetics, 2022, doi:10.3390/biomimetics7040248_

Round 1
Reviewer 1 Report
The authors have presented an interesting development of a new material with antibacterial and underwater sound absorption characteristics. The topic is aligned with the journal, and the paper is, in general well-structured. There are some aspects which must be improved to ensure that the manuscript reaches the quality for the journal. Following, I include the drawbacks detected:
In the introduction, the paper's aim and the proposed material's novelty must be highlighted. Consider using bullet points.
Some Figures are too big. Consider reshaping them.
I strongly recommend splitting the results and the discussion (comparison with existing materials) into different sections.
Other aspects that should be included in the discussion are the cost of the proposed material and its durability in the medium (an estimation, if possible) compared with existing materials.
In order to easily compare the results of the proposed material with existing ones, Table 5 adds the results of the antibacterial efficiency of the tested material.
If possible, create a table that summarizes the characteristics of the proposed material and existing ones in the discussion.
The future work must be highlighted in conclusion in an independent paragraph.
Section 5. Declarations should be erased. Check the template for MDPI.
Minor changes:
Avoid using the keywords the exact words already used in the title. Consider changing "Underwater sound absorption" to another that highlights envirnomental-firendly material or non-toxic material…
Include references for sentence lines 88-89.
Author Response
Reviewer #1:
1) In the introduction, the paper's aim and the proposed material's novelty must be highlighted. Consider using bullet points.
Response: Thank you for your suggestion. We have summarized the novelty of the study and added the following information in the revised version.
“The advantages of the novel coatings are as follow.
(1). The coatings have both underwater sound absorption and antifouling properties, which will fill the gap in this filed.
(2). Without adding any environmentally unfriendly fungicide into the sample, we prepare non-toxic sound-absorbing and antifouling coatings.
(3). In this study, we tested the tensile properties and water erosion resistance of the coatings to prove its durability.” (Page 2, Section 1)
2) Some Figures are too big. Consider reshaping them.
Response: We have modified the size of the figures in the revised draft. We hope the content in the revised draft meets the requirements.
3) I strongly recommend splitting the results and the discussion (comparison with existing materials) into different sections.
Other aspects that should be included in the discussion are the cost of the proposed material and its durability in the medium (an estimation, if possible) compared with existing materials.
Response: Thank you for your recommendation. We adjusted the content of the article and added discussion as Section 4. Further, the following information was also added in the revised version.
“The mass loss rate of pure PDMS is 6.92 g·m-2·h-1, while that of PCSi5 is 2.02 g·m-2·h-1. This represents a 70.8% increase in durability of PCSi5 compared to pure PDMS.” (Page 14 Section 4)
4) In order to easily compare the results of the proposed material with existing ones, Table 5 adds the results of the antibacterial efficiency of the tested material.
If possible, create a table that summarizes the characteristics of the proposed material and existing ones in the discussion.
Response: Thank you very much for pointing this out. We added corresponding characteristics of different materials in Table 5. (Page15 Section 4)
5) The future work must be highlighted in conclusion in an independent paragraph.
Response: We have added the following information in the revised version.
“In this study, the tensile properties of the coatings can be further improved. Therefore, we plan to find new methods to improve the tensile properties of the coating in future work. Optimization of the poor mechanical properties of nanocomposite coatings is a promising work. In addition, we noticed that the antibacterial rate of the coatings can also be improved, and we plan to use more nanofillers in future work to further improve the antibacterial rate of the coatings.” (Page 15 Section 5)
6) Section 5. Declarations should be erased. Check the template for MDPI.
Response: We have checked the template for MDPI and erased Section 5. The following information is added in the revised version.
“Author Contributions: Data curation, P. C., H. W. and M. Z.; Formal analysis, H. W.; Funding acquisition, P. C. and Y. F.; Investigation, H. W. and M. Z.; Methodology, P. C. and Y. F.; Project administration, P. C.; Resources, P. C. and Y. F.; Supervision, P. C. and Y. F.; Validation, H. W. and M. Z.; Visualization, H. W. and M. Z.; Writing - original draft, P. C., H. W. and Y. F.; Writing - review & editing, C. Y. All authors have read and agreed to the published version of the manuscript.
Funding: This research was funded by the National Natural Science Foundation of China, grant number No. 51905468 and 52205197; the financial support of the Natural Science Foundation of Jiangsu Province, grant number No. BK20220551; the ‘Blue Project’ of Yangzhou University, grant number YZU201801 and Senior Talents Research Start Foundation of Jiangsu University, grant number No.5501120017.
Conflicts of interest: The authors declare that they have no known competing financial interests or personal relationships that could have appeared to influence the work reported in this paper.” (Page 16)
7) Avoid using the keywords the exact words already used in the title. Consider changing "Underwater sound absorption" to another that highlights environmental-friendly material or non-toxic material.
Response: Thank you very much for pointing this out. The keyword has been replaced by “environmentally friendly coatings” in the revised version. (Page 1, Keywords)
8) Include references for sentence lines 88-89.
Response: Thank you very much for pointing this out. We have checked this sentence and added references.
“As far as we know, only a few studies have been reported to improve the sound absorption materials’ antifouling properties [22]” (Page 2 Section 1)
References
[22] Huang, Z.S.; Quan Y.Y.; Mao J.J.; Wang Y.L.; Lai Y.K.; Zheng J.; Chen Z.; Wei, K.; Li, H.Q. Multifunctional superhydrophobic composite materials with remarkable mechanochemical robustness, stain repellency, oil-water sepa-ration and sound-absorption properties. Chem. Eng. J., 2019, 358,1610-1619.
https://doi.org/10.1016/j.cej.2018.10.123

Reviewer 2 Report
The authors report a kind of antifouling and sound absorption coating based on PDMS/MWCNT/SiO2. This is a promising method to solve the biofouling of marine application. Therefore, in my opinion, this manuscript should be published in biomimetics after minor revision. For brevity, only major concerns are listed below.
1. For the title, ‘coatings’ may be more suitable than ‘nanocomposites’.
2. In this article, many dates are lack of error bar, such as Figure 7 and Figure 9. The parallel experiments are necessary.
3. The format of Table 5 is incorrect and the presentation is confusing.
4. Some professional words are not used accurately, for example, line 373, ‘pollution release’ can be replaced by ‘fouling release’.
Author Response
Reviewer #2:
(1) For the title, ‘coatings’ may be more suitable than ‘nanocomposites’.
Response: Thank you for your suggestion. The word “nanocomposites” has been replaced by “coatings” in the revised version (Page 1, Title).
(2) In this article, many dates are lack of error bar, such as Figure 7 and Figure 9. The parallel experiments are necessary.
Response: Thank you for pointing this out. We added the necessary experimental data and drew the error bar. The modified figures were included in the revised version.
(3) The format of Table 5 is incorrect and the presentation is confusing.
Response: Thank you very much for your reminding. After checking the template of MDPI, we have modified the format of all tables to make them conform to the standard. We hope that the revised manuscript is correct.
We compared the antifouling performance of this study and the other two studies in the Table 5, including the components of the samples, the bacteria used, and the antibacterial rate. We adjusted the contents of Table 5 in the revised version. (Page 15 Section 5)
(4) Some professional words are not used accurately, for example, line 373, ‘pollution release’ can be replaced by ‘fouling release’.
Response: Thank you for pointing this out. It has been replaced by “fouling release” in the revised version.
